# ANTI-CONCENTRATED CONFIDENCE BONUSES FOR SCALABLE EXPLORATION

**Jordan T. Ash**
Microsoft Research NYC

**Cyril Zhang**
Microsoft Research NYC

**Surbhi Goel**
Microsoft Research NYC

**Akshay Krishnamurthy**
Microsoft Research NYC

**Sham Kakade**
Microsoft Research NYC
Harvard University

## ABSTRACT

Intrinsic rewards play a central role in handling the exploration-exploitation trade-off when designing sequential decision-making algorithms, in both foundational theory and state-of-the-art deep reinforcement learning. The LinUCB algorithm, a centerpiece of the stochastic linear bandits literature, prescribes an elliptical bonus which addresses the challenge of leveraging shared information in large action spaces. This bonus scheme cannot be directly transferred to high-dimensional exploration problems, however, due to the computational cost of maintaining the inverse covariance matrix of action features. We introduce *anti-concentrated confidence bounds* for efficiently approximating the elliptical bonus, using an ensemble of regressors trained to predict random noise from policy network-derived features. Using this approximation, we obtain stochastic linear bandit algorithms which obtain $\tilde{O}(d\sqrt{T})$ regret bounds for $\mathrm{poly}(d)$ fixed actions. We develop a practical variant for deep reinforcement learning that is competitive with contemporary intrinsic reward heuristics on Atari benchmarks.

## 1 INTRODUCTION

Optimism in the face of uncertainty (OFU) is a ubiquitous algorithmic principle for online decision-making in bandit and reinforcement learning problems. Broadly, optimistic decision-making algorithms augment their reward models with a *bonus* (or *intrinsic reward*) proportional to their uncertainty about an action's outcome, ideally balancing exploration and exploitation. A vast literature is dedicated to developing and analyzing the theoretical guarantees of these algorithms (Lattimore & Szepesvári, 2020). In fundamental settings such as stochastic multi-armed and linear bandits, optimistic algorithms are known to enjoy minimax-optimal regret bounds.

In modern deep reinforcement learning, many approaches to exploration have been developed with the same principle of optimism, with most empirical successes coming from uncertainty-based intrinsic reward modules (Burda et al., 2018b; Pathak et al., 2017; Osband et al., 2016). Such bonuses can be very useful, with prior work demonstrating impressive results on a wide array of challenging exploration problems. Several of these methods draw inspiration from theoretical work on multi-armed bandits, using ideas like count-based exploration bonuses. However, a related body of work on *linear* bandits provides tools for extending exploration bonuses to large but structured action spaces, a paradigm which may be appropriate for deep reinforcement learning.

The Linear UCB (LinUCB) algorithm (Auer, 2002; Dani et al., 2008; Li et al., 2010; Abbasi-Yadkori et al., 2011) is attractive in this setting because it enjoys minimax-optimal statistical guarantees. To obtain these, LinUCB leverages a so-called elliptical bonus, the computation of which requires maintaining an inverse covariance matrix over action features. The principal challenge in generalizing the elliptical potential used in bandits to the deep setting lies in computing and storing this object. Due to the moving internal representation of the policy network, and the number of parameters used to compose it, a naive implementation of the LinUCB algorithm would require remembering all of the agent's experience and constantly recomputing and inverting this matrix, which is likely extremely large. Clearly, such an approach is too computationally intensive to be useful. As we discuss in the

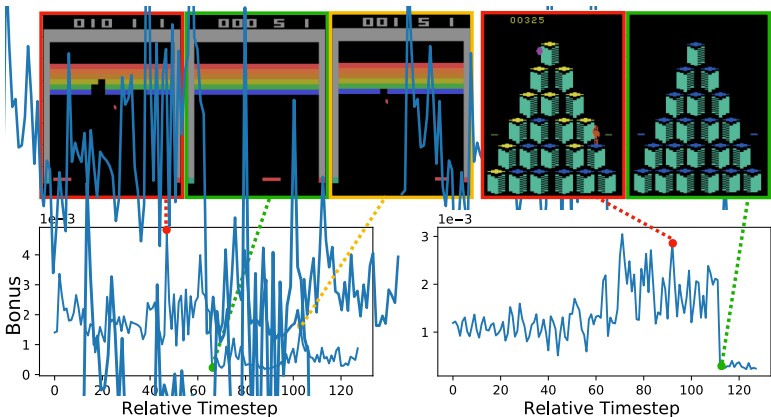

Figure 1: A visualization of the ACB bonus for Atari games Breakout and Q*bert. Large bonuses often correspond to the visitation of states on the periphery of the agent's current experience, for example upon breaking a novel combination of blocks in Breakout or immediately before Q*bert arrives on an unlit platform. When the agent dies, and the game is reset to a familiar state, intrinsic reward drops precipitously.

next section, several works have used neural features for LinUCB-style bonuses, but all require an inversion of this sort, limiting their ability to scale to high dimensions.

Towards bridging foundational algorithms with empirical frontiers, we develop a scalable strategy for computing LinUCB's elliptical bonus, enabling us to investigate its effectiveness as an intrinsic reward in deep reinforcement learning. We use an ensemble of least-squares regressors to approximate these bonuses without explicitly maintaining the covariance matrix or its inverse. Our algorithm is both theoretically principled and computationally tractable, and we demonstrate that its empirical performance on Atari games is often competitive with popular baselines.

## 1.1 OUR CONTRIBUTIONS

We propose the use of *anti-concentrated confidence bounds* (ACB) to efficiently approximate the LinUCB bonus. ACB estimates per-action elliptical confidence intervals by regressing random targets on policy features. It anti-concentrates these bonuses by taking a maximum over the predictions from an ensemble of these regressors.

First, we introduce ACB in the basic stochastic linear bandit setting. We show that these bonuses provably approximate LinUCB's elliptical potential; thus, optimistic exploration with ACB directly inherits standard analysis techniques for LinUCB. We derive near-optimal high-probability regret bounds for ACB, when the size of the action space is polynomial in the action feature dimension. We derive sufficient ensemble sizes for the special cases of multi-armed bandits and fixed actions, as well as the general case of changing actions. These follow from lower tail bounds for the maximum of independent Gaussians; we conjecture that they are improvable using more sophisticated analytical tools.

The main contribution of this work is empirical: we find that ACB provides a viable exploration bonus for deep reinforcement learning. After defining a suitable nonlinear analogue using action features from the policy network, we demonstrate that the intrinsic rewards produced by ACB are competitive with those from state-of-the-art algorithms in deep RL on a variety of Atari benchmarks (Figure 1). To the best of our knowledge, our work is the first to scalably study bonuses from the linear bandits literature in these deep reinforcement learning settings.

## 1.2 RELATED WORK

**Linear bandits.** Stochastic linear bandits were first introduced by Abe & Long (1999). Optimistic algorithms are fundamental in this setting (Auer, 2002; Dani et al., 2008; Li et al., 2010; Rusmevichientong & Tsitsiklis, 2010; Abbasi-Yadkori et al., 2011), and provide minimax-optimal regret bounds.

Several works are concerned with designing more scalable optimism-based algorithms, with a focus on empirical bandit settings (as opposed to deep RL). Jun et al. (2017) consider streaming confidence bound estimates (obtaining per-iteration update costs independent of $t$) in the generalized linear bandits model, but still incur the cost of maintaining the inverse covariance matrix via online Newton step updates. Ding et al. (2021) give strong regret guarantees for epoch-batched SGD with Thompson sampling, under more stringent assumptions (i.i.d. action features, and a "diversity" condition to induce strong convexity). Korda et al. (2015) demonstrate that variance-reduced algorithms succeed in practice for estimating the LinUCB bonus.

Most closely related to our work, Kveton et al. (2019); Ishfaq et al. (2021) study perturbed regressions for exploration, obtaining similar regret bounds to ours (including the suboptimal $\sqrt{\log A}$ factor in the regret bounds). The LinPHE algorithm in (Kveton et al., 2019) uses a single random regressor's constant-probability anti-concentration to approximate the elliptical bonus. In work concurrent to ours, the main algorithm proposed by Ishfaq et al. (2021) also takes a maximum over random regressors, and plugs their analysis into downstream end-to-end results for linear MDPs. The algorithms presented in these works resemble the "always-rerandomizing" variant of ACB (Algorithm 1), which is still not suitable for large-scale deep RL. Our work introduces the "lazily-rerandomizing" and "never-rerandomizing" variants in an effort to close this gap, and presents corresponding open theoretical problems.

Unlike the aforementioned works, we consider our primary contribution as empirical, with the deep learning variant of ACB (Algorithm 2) offering performance comparable to commonly used deep RL bonuses on a wide array of Atari benchmarks without sacrificing theoretical transparency. The resulting algorithmic choices deviate somewhat from the bandit version (as well as LinPHE/LSVI-PHE), towards making ACB a viable drop-in replacement for typical bonuses used in deep RL.

The use of ensembles in this work bears a resemblance to Monte Carlo techniques for posterior sampling (Thompson, 1933; Agrawal & Goyal, 2013). This is a starting point for many empirically-motivated exploration algorithms, including for deep RL (see below), but theory is limited. (Lu & Van Roy, 2017) analyze an ensemble-based approximation of Thompson sampling (without considering a max over the ensemble), and obtain a suboptimal regret bound scaling with $A \log A$.

**Exploration in deep reinforcement learning.**    In deep reinforcement learning, most popular approaches use predictability as a surrogate for familiarity. The Intrinsic Curiosity Module (ICM), for example, does this by training various neural machinery to predict proceeding states from current state-action pairs (Pathak et al., 2017). The $L_2$ error of this prediction is used as a bonus signal for the agent. The approach can be viewed as a modification to Intelligent Adaptive Curiosity, but relying on a representation that is trained to encourage retaining only causal information in transition dynamics (Oudeyer et al., 2007). Stadie et al. (2015) instead use a representation learned by an autoencoder. A large-scale study of these "curiosity-based" approaches can be found in Burda et al. (2018a).

Similar to these algorithms, Random Network Distillation (RND) relies on the inductive bias of a random and fixed network, where the exploration bonus is computed as the prediction error between this random network and a separate model trained to mimic its outputs (Burda et al., 2018b).

In tabular settings, count-based bonuses are often used to encourage exploration (Strehl & Littman, 2008). This general approach has been extended to the deep setting as well, where states might be high-dimensional and infrequently visited (Bellemare et al., 2016). Algorithms of this type usually rely on some notion of density estimation to group similar states (Ostrovski et al., 2017; Tang et al., 2016; Martin et al., 2017).

Other work proposes more traditional uncertainty metrics to guide the exploration process. For example, Variational Intrinsic Control (Gregor et al., 2016) uses a variational notion of uncertainty while Exploration via Disagreement (Pathak et al., 2019) and Bootstrapped DQN  (Osband et al., 2016) model uncertainty over the predictive variance of an ensemble. More broadly, various approximations of Thompson sampling (Thompson, 1933) have been used to encourage exploration in deep reinforcement learning (Osband et al., 2013; Guez et al., 2012; Zhang et al., 2020; Strens, 2000; Henaff et al., 2019). These methods have analogues in the related space of active learning, where ensemble and variational estimates of uncertainty are widely used (Gal et al., 2017; Beluch et al., 2018).

Separate from these, there has been some work intended to more-directly deliver UCB-inspired bonuses to reinforcement learning (Zhou et al., 2020; Nabati et al., 2021; Zahavy & Mannor,

2019; Bai et al., 2021). These efforts have various drawbacks, however, with many requiring matrix inversions, limiting their ability to scale, or simply not being competitive with more conventional deep RL approaches. ACB removes the covariance matrix inversion requirement, and as we demonstrate in Section A.2, is competitive with RND and ICM on several Atari benchmarks.

## 2 PRELIMINARIES

In this section, we provide a brief review and establish notation for the stochastic linear bandit setting and the LinUCB algorithm. For a more comprehensive treatment, see Agarwal et al. (2019); Lattimore & Szepesvári (2020). At round $t = 1, \ldots, T$, the learner is given an action set $\mathcal{A}_t \subseteq \mathcal{A}$ with features $\{x_{t,a} \in \mathbb{R}^d\}_{a \in \mathcal{A}_t}$, chooses an action $a_t \in \mathcal{A}_t$, and receives reward $r_t(a_t) := \langle x_{t,a_t}, \theta^* \rangle + \varepsilon_t$, where $\varepsilon_t$ is a martingale difference sequence. We use $A$ to denote $\max_t |\mathcal{A}_t|$. The performance of the learner is quantified by its regret,

$$R(T) = \sum_{t=1}^{T} \left\langle x_{t,a_t^*}, \theta^* \right\rangle - \left\langle x_{t,a_t}, \theta^* \right\rangle, \quad a_t^* := \arg\max_{a \in \mathcal{A}_t} \left\langle x_{t,a}, \theta^* \right\rangle.$$

This setting encompasses two commonly-considered special cases:

- *Fixed actions:* the action sets $\mathcal{A}_t$ are the same for all $t$.
- *Multi-armed bandits:* fixed actions, with $\mathcal{A}_t = [d]$ and $x_{t,a} = e_a$. Here, $\theta^*$ is the vector of per-arm means.

An OFU-based algorithm plays at round $t$

$$a_t := \arg\max_{a \in \mathcal{A}_t} \left\{ \hat{r}_t(a) + \text{bonus}_t(a) \right\},$$

where $\hat{r}_t(a)$ is an estimate for the reward $r_t(a)$, and $\text{bonus}_t(a)$ is selected to be a valid upper confidence bound for $\hat{r}_t(a)$, such that $\mathbb{E}[r_t(a)] \leq \hat{r}_t(a) + \text{bonus}_t(a)$ with high probability. The LinUCB algorithm uses $\hat{r}_t(a) := \left\langle x_{t,a}, \hat{\theta}_t \right\rangle$, where $\hat{\theta}_t = \arg\min_{\theta \in \mathbb{R}^d} \sum_{\tau=1}^{t-1} \left( \langle x_{\tau,a_\tau}, \theta \rangle - r_i(a_i) \right)^2 + \lambda \|\theta\|^2$ is the $2$-regularized least-squares regressor for the reward, and sets $\text{bonus}_t(a) = \beta \sqrt{x_{t,a}^\top \tilde{\Sigma}_t^{-1} x_{t,a}}$, where $\tilde{\Sigma}_t := \lambda I + \sum_{\tau=1}^{t-1} x_{t,a_t} x_{t,a_t}^\top$. In an online learning context, this is known as the elliptical potential (Cesa-Bianchi & Lugosi, 2006; Carpentier et al., 2020). We will quantify the theoretical performance of ACB by comparing it to LinUCB. Theorem 1 gives a high-probability regret bound for LinUCB, which is optimal up to logarithmic factors:

**Theorem 1** (LinUCB regret bound; Theorem 5.3, (Agarwal et al., 2019)). *If $\|\theta^*\|, \|x_{t,a}\| \leq O(1)$, and each $\varepsilon_t$ is $O(1)$-subgaussian, then there are choices of $\lambda, \beta$ such that, with probability at least $1 - \delta$, the regret of the LinUCB algorithm satisfies*

$$R(T) \leq \tilde{O}\left( d\sqrt{T} \right).$$

*The $\tilde{O}(\cdot)$ suppresses factors polynomial in $\log(1/\delta)$ and $\log T$.*

LinUCB can be implemented in $O(d^2|\mathcal{A}_t|)$ time per iteration, by maintaining $\tilde{\Sigma}_t^{-1}$ using rank-1 updates, and using this to compute the least-squares estimator $\hat{\theta}_t = \tilde{\Sigma}_t^{-1} \sum_{\tau=1}^{t-1} x_{\tau,a_\tau} r_\tau(a_\tau)$ as well as $\text{bonus}_t(a)$.

## 3 ANTI-CONCENTRATED CONFIDENCE BOUNDS FOR LINEAR BANDITS

We begin by presenting the ACB intrinsic reward for the stochastic linear bandit setting. It is well-known that the weights of a least-squares regressor trained to predict i.i.d. noise will capture information about about the precision matrix of the data. Our key idea is to amplify the deviation in the predictions of an ensemble of such regressors by taking a maximum over their outputs, allowing us to construct upper confidence bounds for $\hat{r}_t(a)$. ACB maintains $M$ linear regressors trained to

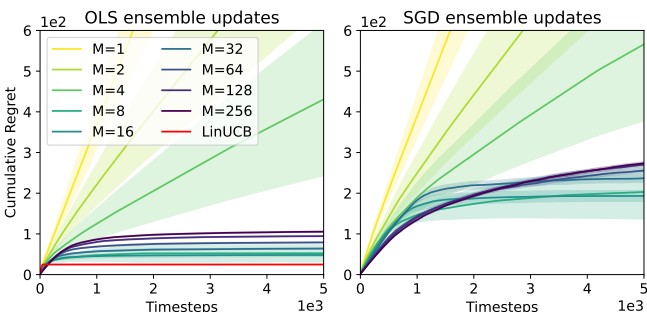

Figure 2: The cumulative regret for ACB (incremental sampling) with different ensemble sizes. We show a multi-armed bandit setting with 50 actions, comparing exact ordinary least squares (left) with SGD using Polyak averaging (right). Up to a point, increasing ensemble size improves cumulative regret in both cases.

---

**Algorithm 1** ACB for linear bandits

---

**Parameters:** ensemble size $M$; $\beta, \lambda$

1: For all $i = 1, \ldots, d$, set warm-start features $x_{-i+1} = \sqrt{\lambda} e_i$ and corresponding warm-start targets $y_{-i+1}^{(j)} \sim \mathcal{N}(0,1)$ for each $j \in [M]$

2: **for** $t = 1, \ldots, T$: **do**

3:     Observe action features $\{x_{t,a}\}_{a \in \mathcal{A}_t}$

4:     Update reward estimator:
$$\hat{\theta}_t := \arg\min_{\theta} \sum_{\tau=1}^{t-1} \left( \langle x_{\tau,a_\tau}, \theta \rangle - r_\tau(a_\tau) \right)^2 + \lambda \|\theta\|^2$$

5:     Sample new random ensemble targets:

     *(re-randomized)*     $y_\tau^{(j)} \sim \mathcal{N}(0,1)$ for each $j \in [M], \tau \in \{-d+1, \ldots, t-1\}$

     *(incremental)*     $y_{t-1}^{(j)} \sim \mathcal{N}(0,1)$ for each $j \in [M]$

6:     Update bonus ensembles for each $j \in [M]$,
$$w_t^{(j)} := \arg\min_{w} \sum_{\tau=-d+1}^{t-1} \left( \langle x_{\tau,a_\tau}, w \rangle - y_\tau^{(j)} \right)^2$$

7:     Compute per-arm bonuses $\mathrm{bonus}_t(a) := \beta \cdot \max_{j \in [M]} \left| \left\langle x_{t,a}, w_t^{(j)} \right\rangle \right|$ for each $a \in \mathcal{A}_t$

8:     Choose action $a_t := \arg\max_{a \in \mathcal{A}_t} \langle x_{t,a}, \hat{\theta}_t \rangle + \mathrm{bonus}_t(a)$

9:     Observe reward $r_t(a_t) = \langle x_{t,a_t}, \theta^* \rangle + \varepsilon_t$

---

predict i.i.d. standard Gaussian noise, and sets the bonus to be proportional to the maximum deviation over this ensemble from its mean (which is 0). This procedure is formally provided in Algorithm 1.

With the correct choice of ensemble size, the concentration of the maximum of independent Gaussians implies that each $\mathrm{bonus}_t(a)$ is large enough to be a valid upper confidence bound, yet small enough to inherit the regret bound of LinUCB. ACB can be instantiated with freshly resampled history; in this case, the theoretical guarantees on the sufficient ensemble size are strongest. A more computationally efficient variant, which is the one we scale up to deep RL, samples random targets only once, enabling the use of streaming algorithms to estimate regression weight $w_t^j$. For these two variants, the following regret bounds result from straightforward modifications to the analysis of LinUCB (Abbasi-Yadkori et al., 2011), which are deferred (along with more precise statements) to Appendix B:

**Proposition 2** (ACB regret bound with re-randomized history). *Then, Algorithm 1, with re-randomized history and ensemble size $M = \Theta(\log(T/\delta))$, obtains a regret bound of $\tilde{O}(d\sqrt{T \log A})$, with probability at least $1 - \delta$.*

For $A \leq O(\mathrm{poly}(d))$, our regret bound matches that of LinUCB up to logarithmic factors. We note that the expected regret guarantee of the LinPHE algorithm in Kveton et al. (2019) also has the same dependence on $\log(AT)$. Our lower-tail analysis on the number of regressors required for the maximum deviation to be optimistic is slightly sharper than that found in Ishfaq et al. (2021), by a factor of $\log A$; this is due to requiring optimism only on $a^*$ in Lemma 2.

---

**Algorithm 2** ACB exploration for reinforcement learning

---

**Parameters:** ensemble $M$, tail-average constant $\gamma$, policy network function $f$ parameterized by $\theta$, update frequency $\tau$

1: Initialize auxiliary weights $w_0^{(j)} \in \mathbb{R}^d, \forall j \in [M]$
2: Initialize auxiliary policy network parameters $\theta_{\mathsf{aux}} \leftarrow \theta$
3: **for** $t = 1, \dots, T$: **do**
4:     Observe state $x_t$
5:     Compute gradient features using auxiliary policy network $g_t := g(x_t; f, \theta_{\mathsf{aux}})$
6:     Evaluate bonus $b_t = \max_j (((w_t^{(j)})^\top g_t)^2)$
7:     Select action from policy network softmax distribution $f(x_t; \theta)$
8:     **if** $t$ is a multiple of $\tau$ **then**
9:         Update policy network parameters $\theta$ with PPO on intrinsic rewards $\{b_{t-\tau+1}, \dots, b_t\}$
10:        For all $j \in [M]$, draw random targets $y_t^{(j)} \sim \mathcal{N}(0, 1)$ and compute loss

$$L_t^{(j)} = ((w_t^{(j)})^\top g_t - y_t^{(j)})^2 + \lambda ||w_t^{(j)} - w_0^{(j)}||_2^2$$

11:        For all $j \in [M]$, update $w_{t+1}^{(j)}$ using $L_t^{(j)}$                 //*using RMSProp*
12:        Update auxillary network parameters $\theta_{\mathsf{aux}} \leftarrow \alpha\theta + (1-\alpha)\theta_{\mathsf{aux}}$

---

**Proposition 3** (ACB regret bound with incremental updates). *Algorithm 1, with incrementally randomized history and an ensemble size of $M = \Theta(\log(T/\delta))$, obtains a regret bound of $\tilde{O}(A\sqrt{T})$ in the multi-armed bandit setting, with probability at least $1 - \delta$.*

We conjecture that the incrementally-updated version guarantees can be extended to linear bandits beyond MAB for reasonable ensemble sizes. In between the extremes of always vs. never re-randomizing history, it is possible to analyze a "rarely re-randomizing" variant of the algorithm, which obtains the smaller sufficient ensemble size of Proposition 2 while only re-randomizing $\tilde{O}(d \log T)$ times:

**Theorem 4** (Lazy re-randomization). *Algorithm 3, with lazy re-randomized history, obtains a regret bound of $\tilde{O}(d\sqrt{\log(AT/\delta)T})$ in the case of linear bandits with fixed actions, while resampling the random targets $y_\tau^{(j)}$ only $\tilde{O}(d \log(AT/\delta) \log(T))$ times.*

We defer the modified algorithm and analysis to Appendix C. We note that the lazy version is computationally more efficient than the bonus approximation procedures in LinPHE (Kveton et al., 2019) and LSVI-PHE (Ishfaq et al., 2021), and the expected regret guarantees obtained by LinPHE are not sufficient to successfully implement a lazy strategy.

Like other works proposing more scalable algorithms in the same setting (Korda et al., 2015; Ding et al., 2021), with theoretically sound hyperparameter choices, we do not exhibit end-to-end theoretical improvements over the $O(d^2|\mathcal{A}_t|)$ per-iteration cost of rank-1 updates for implementing LinUCB. Both lines 4 and 6 in Algorithm 1 depend on *recursive least squares* optimization oracles. For our main empirical results (as well as Figure 2 (right)), we implement this oracle using Polyak-averaged SGD updates to each regressor, requiring $O(Md|\mathcal{A}_t|)$ time per iteration to compute all the bonuses.

**Synthetic bandit experiments.** As a simple empirical validation of our method, we demonstrate the performance of Algorithm 1 (with incremental sampling) in a simulated multi-armed bandit setting with $A = 50$ actions. One arm has a mean reward 0.75 while the others' are set to 0.25; the noise $\varepsilon_t$ for each arm is sampled independently from $\mathcal{N}(0, 0.1^2)$. As predicted by the theory, the ensemble size $M$ must be chosen to be large enough for the bonuses to be sufficiently optimistic, small enough to avoid too many upper-tail events. Further details are provided in Appendix A.1.

## 4 EXTENDING ACB TO DEEP REINFORCEMENT LEARNING

Outlined as Algorithm 2, the RL version of ACB deviates from the version mentioned earlier in several ways. First, like other work assigning exploratory bonuses in deep reinforcement learning, we assign intrinsic rewards on states, rather than on state-action pairs (Pathak et al., 2017; Burda et al., 2018b).

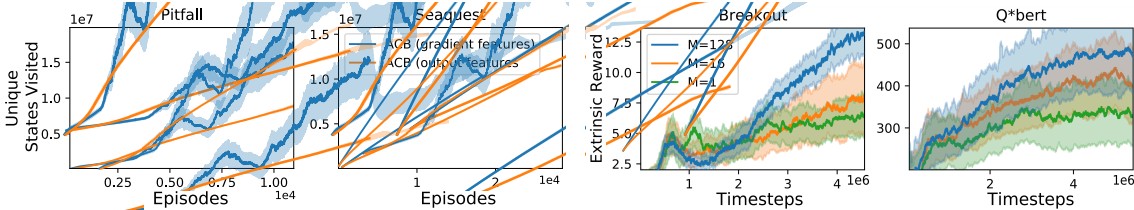

Figure 3: ACB with features using either penulti-mate layer or gradient features. Overall, gradient features seem to offer improved exploration, as measured by the number of unique states visited by an agent trained exclusively on these bonuses.

Figure 4: Larger ACB ensembles are generally more stable, resulting in better exploration. An agent trained exclusively on large-ensemble ACB bonuses will tend to observe more extrinsic re-ward than those with fewer regressors.

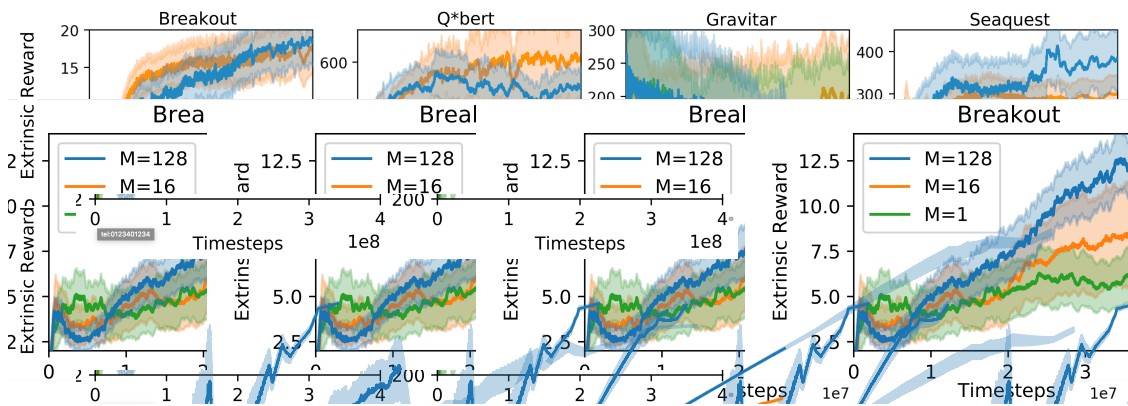

Figure 5: Extrinsic reward in Atari games as a function of the number of times the agent interacts with the environment. Each plot corresponds to a different intrinsic reward scheme, and PPO is being trained only to maximize this intrinsic reward.

Second, to help deal with non-stationarity, we compute gradient features with respect to a tail-averaged set of policy weights, rather than on the policy weights being used to interact with the environment.

Because neural networks learn their own internal feature representation, extending ACB to the neural setting gives us some flexibility in the way we choose to represent data. One option is to use the penultimate layer representation of the network, an approach that has seen success in the closely related active learning setting (Sener & Savarese, 2018). In the deep active learning classification setting, a more recent approach is to instead represent data using a hallucinated gradient, embedding data as the gradient that would be induced by the sample if the most likely label according to the model were the true label (Ash et al., 2020). Treating the policy network as a classifier, we find that this representation tends to perform better than using just the penultimate layer. Specifically, this gradient is computed as

$$g(x; f, \theta) = \frac{\partial}{\partial \theta} \ell_{\mathsf{ce}}(f(x; \theta), \hat{y}), \quad \hat{y} = \arg\max f(x; \theta), \tag{1}$$

where $f$ is the policy network, $\theta$ is its current parameters, $\ell_{\mathsf{ce}}(\tilde{y}, \hat{y})$ is the log loss of the pre-diction $\tilde{y}$ given true label $\hat{y}$, and $\hat{y}$ is the action to which the most probability mass is assigned. Using this representation, each seen state $x$ contributes $g_x g_x^\top$ to the ACB covariance matrix, a quantity that can viewed as a sort of rank-one approximation of the pointwise Fisher information, $I(x, \theta) = \mathbb{E}_{y \sim f(x, \theta)} \nabla \ell(f(x, \theta), y)(\nabla \ell(f(x, \theta), y))^\top$ (Ash et al., 2021). This connection draws a parallel between ACB with gradient features and well-understood, Fisher-based objectives in the active learning literature (Zhang & Oles, 2000; Gu et al., 2014; Chaudhuri et al., 2015). The ability to use such a gradient representation highlights the scalability of ACB—policy networks used in RL are typically composed of over a million parameters, and explicitly maintaining and inverting a covariance

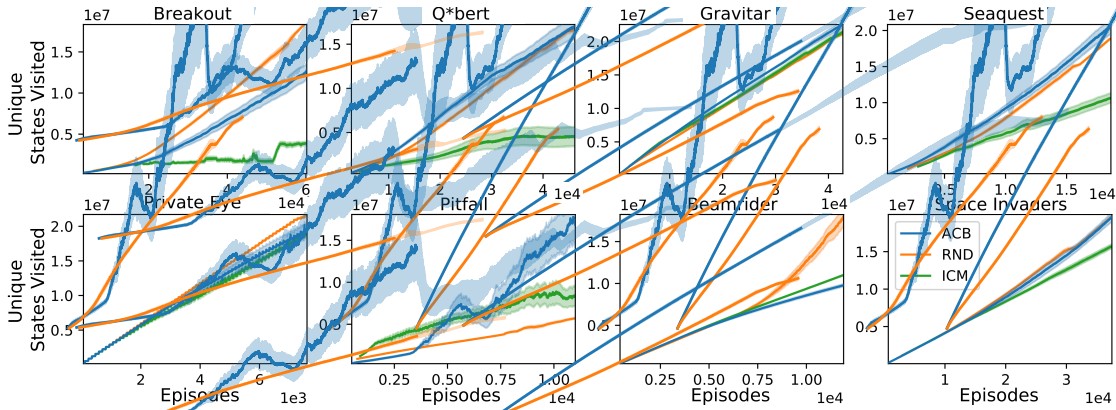

Figure 6: The number of unique states visited (measured by a hashing) in Atari games as a function of the number of game episodes. Each plot corresponds to a different intrinsic reward scheme, and PPO is being trained only to maximize this intrinsic reward.

matrix of this size is not feasible. ACB circumvents this need and handles high-dimensional features without issue. Figure 3 shows examples of environments for which the gradient representation produces better exploration than the penultimate-layer representation, according to the number of unique states encountered by the agent (using a hash). In both cases, representations are computed on a Polyak-averaged version of the policy with $\alpha = 10^{-6}$, which we also use in Section 5 below.

## 5 EXPERIMENTS

This section compares ACB with state-of-the-art approaches to exploration bonuses in deep reinforcement learning on Atari benchmarks. We inherit hyperparameters from RND, including the convolutional policy network architecture, which can be found in Appendix Section A.2. Policy optimization is done with PPO (Schulman et al., 2017). In all experiments, we use 128 parallel agents and rollouts of 128 timesteps, making $\tau$ in Algorithm 2 $128^2$. Like other work on intrinsic bonuses, we consider the non-episodic setting, so the agent is unaware of when it the episode is terminated. Consequently, even in games for which merely learning to survive is equivalent to learning to play well, random or constant rewards will not elicit good performance. All experiments are run for five replicates, and plots show mean behavior and standard error.

In our investigation, we identified several tricks that seem to help mitigate the domain shift in the gradient representation. Among these, as mentioned above, we tail-average auxiliary policy network parameters, slowly moving them towards the deployment policy. This approach has been used to increase stability in both reinforcement learning and generative adversarial networks (Lillicrap et al., 2015; Bell, 1934). In experiments shown here, $\alpha$ is fixed at $10^{-6}$. We additionally normalize intrinsic bonuses by a running estimate of their standard deviation, an implementation detail that is also found in both RND and ICM. Batch normalization is applied to gradient features before passing them to the auxiliary weight ensemble, and, like in the bandit setting, we compute bonuses using a Polyak-averaged version of the ensemble rather than the ensemble itself. We use the RMSprop variant of SGD in optimizing auxiliary weights.

We use an ensemble of 128 auxiliary weights and regularize aggressively towards their initialization with $\lambda = 10^3$. As with the bandit setting, larger ensembles tend to produce bonuses that both are more stable and elicit more favorable behavior from the agent (Figure 4). Experiments here consider eight different Atari games. Environments were chosen to include both games that are known to be challenging in the way of exploration (Private Eye, Pitfall! and Gravitar (Burda et al., 2018b)), and games that are not—remaining environments overlap with those from Table 1 of Mnih et al. (2015))

**Intrinsic rewards only.** In our first set of experiments, we train agents exclusively on bonuses generated by ACB, rather than on any extrinsic reward signal. In Figure 1, we visualize some of these bonuses as gameplay progresses in two Atari games. In both, large intrinsic bonuses are obtained when the agent reaches the periphery of its experience. Correspondingly, when the agent dies, and the game state is reset to something familiar, the ACB bonus typically drops by a significant margin.

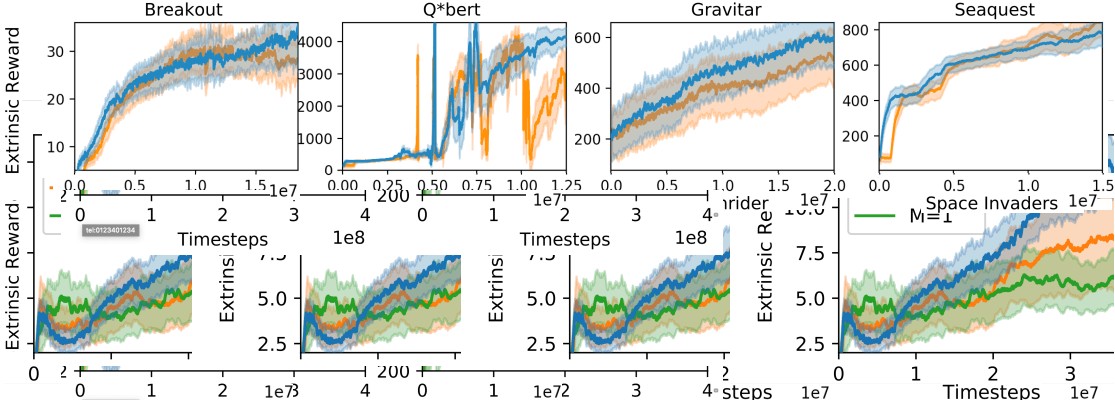

Figure 7: Extrinsic reward in Atari games as a function of the number of frames seen by the agent. Each plot corresponds to a different intrinsic reward scheme, and PPO is being trained to jointly maximize these bonuses and the observed extrinsic reward. ACB is competitive with RND in this setting, and for some environments and experience budgets, even obtains higher reward.

These intrinsic bonuses encourage the agent to avoid the start state, and to continually push the boundary of what has been observed so far. Figure 5 compares the per-episode extrinsic rewards collected by agents trained only on intrinsic rewards across eight popular Atari games and three different bonus schemes. Even though extrinsic reward is not observed, many of these agents obtain non-trivial performance on the game, suggesting that the pursuit of novelty is often enough to play well. Extrinsic reward does not necessarily correlate with exploration quality, but ACB is clearly competitive with other intrinsic reward methods along this axis. In Figure 6, we measure the number of unique states encountered, as measured by a hash, as exploration progresses. ACB is competitive from this perspective as well, visiting as many or more states than baselines in most of the environments considered.

**Intrinsic and extrinsic rewards.** Next, while still in the non-episodic setting, we experiment with training PPO jointly on intrinsic and extrinsic rewards. Following the PPO modification proposed by Burda et al. (2018b), we use two value heads, one for intrinsic and another for extrinsic rewards. total reward is computed as the sum of intrinsic and extrinsic returns. Figure 7 compares RND and ACB agents trained with both intrinsic and extrinsic rewards on eight Atari games, which shows ACB again performing competitively in comparison to RND. We find that ACB performs at least as well as RND on most of the games we considered, and sometimes, for particular timesteps and environments, even offers an improvement.

## 6 CONCLUSION

This article introduced ACB, a scalable and computationally-tractable algorithm for estimating the elliptical potential in both bandit and deep reinforcement learning scenarios. ACB avoids the need to invert a large covariance matrix by instead storing this information in an ensemble of weight vectors trained to predict random noise.

This work aims to bridge the gap between the empirical successes of deep reinforcement learning and the theoretical transparency of bandit algorithms. We believe ACB is a big step along this direction, but there is still more work to be done to further this effort. For one, ACB is still not able to perform as well as RND on Montezuma's Revenge, an Atari benchmark for which exploration is notoriously difficult, and for which RND performs exceptionally well.

Our experiments indicate that our proposed technique of anti-concentrating optimism can improve empirical performance well beyond our current theoretical understanding. We leave improving our regret guarantees (such as eliminating dependences on $\log(A)$, matching the optimal rate in all regimes), as well as extending our analysis to more difficult settings (such as the incremental case) as exciting open problems.

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

## A EXPERIMENTAL DETAILS

### A.1 BANDIT EXPERIMENTS

Figure 2 shows the cumulative regret of Algorithm 1 (with incremental sampling) in a simulated multi-armed bandit setting with $A = 50$ actions, for different values of $M$ (powers of 2 from 1 to 256). One randomly sampled arm is chosen to have a mean reward of $0.75$, with the others $0.25$; the noise $\varepsilon_t$ for each arm is sampled independently from $\mathcal{N}(0, 0.1^2)$. Standard deviations are shown over 100 independent trials. The bonus scaling factor $\beta$ was selected in each run by grid search over $\{0.01, 0.05, 0.1, 0.5, 1.0, 5.0, 10.0\}$; the learning rate for SGD was selected by grid search over the same values. Each batch of experiments (one choice of algorithm and $M$; all replicas and hyperparameter sweeps) took around 1 CPU hour on an internal cluster.

### A.2 ATARI EXPERIMENTS

All policy network architectures used, for ICM, RND, and ACB, have the convolutional architecture described in Burda et al. (2018b), consisting of three convolutional layers followed by two linear layers, and leaky ReLU activations. For ACB and RND, we use PPO hyperparameters also from Burda et al. (2018b), including 4 update epochs, an entropy coefficient of 0.001, $\gamma_E = 0.999$ and $\gamma_I = 0.99$. Our ICM implementation uses the default parameters of Pathak et al. (2017), which instead use 3 update epochs. We use a learning rate of 0.0001 and 128 simultaneous agents universally. All model updates are performed via Adam except for the ACB auxiliary weights, which use RMSprop. See attached code for more details.

Compute resources per job included a single CPU and either a P100 of V100 NVIDIA GPU, with experiments each taking between three and five days.

## B PROOFS OF REGRET BOUNDS

In this section, we prove Propositions 2 and 3. We use the following notation in the analysis:

- $\tilde{\Sigma}_t$ denotes $\lambda I + \sum_{\tau=1}^{t-1} x_{t,a_t} x_{t,a_t}^\top$.
- $\mathrm{regret}_t$ denotes the instantaneous regret $\langle x_{t,a^*}, \theta^* \rangle - \langle x_{t,a_t}, \theta^* \rangle$.
- $\log^+(\cdot)$ denotes $\log(\max(1, \cdot))$.

Furthermore, we assume that $||x_{t,a}|| \le B$ for all $t, a$, $||\theta^*|| \le W$, and that the $\eta_t$ are a martingale difference sequence independent of the decisions made by the algorithm. Assume that the $\eta_t$ are $\sigma^2$-subgaussian, with $\sigma > 0$.

### B.1 USEFUL LEMMAS

**Lemma 5** (Tail bounds for a maximum of Gaussians; Tu (2017)). *Let $z^{(1)}, \ldots, z^{(M)}$ be i.i.d. $\mathcal{N}(0, 1)$ random variables. Let $Z$ denote $\max_{j=1,\ldots,M} |z^{(j)}|$. Then, for all $0 < \delta < 1$, each of these inequalities holds with probability at least $1 - \delta$:*

$$Z \ge \sqrt{\frac{\pi}{2}} \sqrt{\log(M/2) - \log\log(1/\delta)}.$$

$$Z \le \sqrt{2} \left( \sqrt{\log(2M)} + \sqrt{\log(1/\delta)} \right).$$

**Lemma 6** (Proposition 5.5, Agarwal et al. (2019)). *At any time $t \in [T]$, for all $a \in \mathcal{A}_t$, with probability $1 - \delta_t$,*

$$\left\langle a, \theta^* - \hat{\theta}_t \right\rangle \le \beta_t \|a\|_{\tilde{\Sigma}_t^{-1}}$$

*for $\beta_t = \sqrt{\lambda}\|\theta^*\| + \sqrt{2\sigma^2 \log(\det(\tilde{\Sigma}_t)\lambda^{-d}/\delta_t)}$.*

**Lemma 7** (Sufficient conditions for the bonus). *Suppose that for all $t \leq T$, the bonuses satisfy*

$$\text{bonus}_t(a) \leq \gamma_2 \|x_{t,a}\|_{\tilde{\Sigma}_t^{-1}} \qquad\qquad \forall\, a \in \mathcal{A}_t$$

$$\text{bonus}_t(a^*) \geq \gamma_1 \|x_{t,a^*}\|_{\tilde{\Sigma}_t^{-1}}$$

*where $a^*$ is the optimal action. If we choose actions according to this bonus at every time step $t$, then with probability $1 - \delta$,*

$$R(T) \leq (\gamma_2 + \bar{\beta}) \sqrt{dT \log\left(1 + \frac{TB}{\lambda d}\right)}$$

*for $\bar{\beta} = \sqrt{\lambda} W + \sqrt{2}\sigma \sqrt{d \log\left(1 + \frac{TB}{\lambda d}\right) + \log(T/\delta)}$, as long as $\gamma_1 \geq \bar{\beta}$.*

*Proof.* Using the assumptions on the bonus and Lemma 6 with $\delta_t = \delta/T$, [1] with probability $1 - \delta$, for all $t \leq T$

$$\begin{aligned}
\text{regret}_t &= \langle x_{t,a^*}, \theta^* \rangle - \langle x_{t,a_t}, \theta^* \rangle \\
&\leq \left\langle x_{t,a^*}, \hat{\theta}_t \right\rangle + \beta_t \|x_{t,a^*}\|_{\tilde{\Sigma}_t^{-1}} - \langle x_{t,a_t}, \theta^* \rangle \\
&\leq \left\langle x_{t,a_t}, \hat{\theta}_t \right\rangle + \text{bonus}_t(a_t) - \text{bonus}_t(a^*) + \beta_t \|x_{t,a^*}\|_{\tilde{\Sigma}_t^{-1}} - \langle x_{t,a_t}, \theta^* \rangle \\
&\leq \text{bonus}_t(a_t) - \text{bonus}_t(a^*) + \beta_t \|x_{t,a^*}\|_{\tilde{\Sigma}_t^{-1}} + \beta_t \|x_{t,a_t}\|_{\tilde{\Sigma}_t^{-1}} \\
&\leq \gamma_2 \|x_{t,a_t}\|_{\tilde{\Sigma}_t^{-1}} - \gamma_1 \|x_{t,a^*}\|_{\tilde{\Sigma}_t^{-1}} + \beta_t \|x_{t,a^*}\|_{\tilde{\Sigma}_t^{-1}} + \beta_t \|x_{t,a_t}\|_{\tilde{\Sigma}_t^{-1}} \\
&\leq (\gamma_2 + \beta_t) \|x_{t,a_t}\|_{\tilde{\Sigma}_t^{-1}}.
\end{aligned}$$

Summing up, we have

$$\begin{aligned}
R(T) = \sum_{t=1}^{T} \text{regret}_t &\leq \sum_{t=1}^{T} (\gamma_2 + \beta_t) \|x_{t,a_t}\|_{\tilde{\Sigma}_t^{-1}} \\
&\leq \sqrt{\sum_{t=1}^{T} (\gamma_2 + \beta_t)^2} \sqrt{\sum_{t=1}^{T} \|x_{t,a_t}\|_{\tilde{\Sigma}_t^{-1}}^2} \\
&\leq \sqrt{\sum_{t=1}^{T} (\gamma_2 + \beta_t)^2} \sqrt{2 \log \frac{\det(\tilde{\Sigma}_T)}{\det(\lambda I)}} \\
&\leq (\gamma_2 + \bar{\beta}) \sqrt{2dT \log\left(1 + \frac{TB}{\lambda d}\right)}.
\end{aligned}$$

The above follows from Cauchy-Schwarz and observing that $\beta_t \leq \bar{\beta}$ for all $t \leq T$. $\qquad\square$

### B.2 REGRET BOUNDS

**Proposition 2** (ACB regret bound with re-randomized history). *Algorithm 1, with re-randomized history, ensemble size $M = \lceil \log(2T/\delta) \rceil$, $\lambda = \sigma^2/W^2$, and $\beta = \sqrt{\lambda} W + \sqrt{2}\sigma \sqrt{d \log\left(1 + \frac{TB}{\lambda d}\right) + \log(T/\delta)}$, as long as $\gamma_1 \geq \bar{\beta}$, obtains a regret bound of $R(T) \leq O\left(\sigma d \sqrt{T \log A} \log(T/\delta) \log^+\left(\frac{TBW}{\sigma d}\right)\right)$, with probability at least $1 - \delta$.*

*Proof.* At time $t$, since we re-sample all the noisy targets, we have,

$$w_t^{(j)} = \tilde{\Sigma}_t^{-1} \sum_{\tau = -d+1}^{t-1} x_{\tau,a_\tau} y_\tau^{(j)}.$$

---

[1] If we want it to hold for all $t < \infty$, we can choose $\delta_t = \frac{3\delta}{\pi^2 t^2}$. This would give essentially the same bound up to constants.

$\left\langle w_t^{(j)}, x_{t,a} \right\rangle$ is therefore distributed as $\mathcal{N}\left(0, \|x_{t,a}\|_{\tilde{\Sigma}_t^{-1}}^2\right)$. Thus by Lemma 5, for all $a \in \mathcal{A}_t$ with probability $1 - \delta$,

$$\max_{j \in [M]} \left|\left\langle w_t^{(j)}, x_{t,a} \right\rangle\right| \leq c_2 \|x_{t,a}\|_{\tilde{\Sigma}_t^{-1}} \sqrt{\log M + \log(AT/\delta)},$$

and for $a^*$,

$$c_1 \|x_{t,a^*}\|_{\tilde{\Sigma}_t^{-1}} \sqrt{\log M - \log\log(T/\delta)} \leq \max_{j \in [M]} \left|\left\langle w_t^{(j)}, x_{t,a^*} \right\rangle\right|$$

where $c_1, c_2 > 0$ are fixed constants. Since $\mathrm{bonus}_t(a) = \beta \max_{j \in [M]} \left|\left\langle w_t^{(j)}, x_{t,a} \right\rangle\right|$, we get a lower and upper bound on the bonus in terms of the LinUCB bonus. Lastly, by the choice of $M$, with probability $1 - \delta$, we satisfy conditions of Lemma 7 with $c_1 \sqrt{\log 2}\beta = \sqrt{\lambda}W + \sqrt{2}\sigma\sqrt{d\log\left(1 + \frac{TB}{\lambda d}\right) + \log(T/\delta)}$. This gives us

$$R(T) \leq \left(\frac{2c_2\sqrt{\log(AT/\delta)}}{c_1} + 1\right)\left(\sqrt{\lambda}W + \sqrt{2}\sigma\sqrt{d\log\left(1 + \frac{TB}{\lambda d}\right) + \log(T/\delta)}\right)\sqrt{dT\log\left(1 + \frac{TB}{\lambda d}\right)}$$

$$= O\left(\sigma d\sqrt{T\log A}\log(T/\delta)\log^+\left(\frac{TBW}{\sigma d}\right)\right).$$

$\square$

**Proposition 3** (ACB regret bound with incremental updates). *Assume further that the $\eta_t$ are i.i.d. $\sigma^2$-subgaussian random variables. Algorithm 1, with incrementally randomized history, ensemble size $M = \lceil \log(T/\delta) \rceil$, $\lambda = \sigma^2/W^2$, and $\beta = \sqrt{\lambda}W + \sqrt{2}\sigma\sqrt{A\log\left(1 + \frac{TB}{\lambda A}\right) + \log(T/\delta)}$, as long as $\gamma_1 \geq \bar{\beta}$, obtains a regret bound of*

$$R(T) \leq O\left(\sigma A\sqrt{T\log A}\log(T/\delta)\log^+\left(\frac{TBW}{\sigma A}\right)\right)$$

*in the multi-armed bandit setting, with probability at least $1 - \delta$.*

*Proof.* Recall that the action set $\mathcal{A}$ is fixed, and $d = A$; we will use $x_a$ to denote $x_{t,a}$. We follow a "reward tape" argument (see, e.g., Slivkins (2019), Section 1.3.1) to show that the conditions of Lemma 7 hold. Notice that in the incremental version of Algorithm 1, $\mathrm{bonus}_t(a)$ depends only on the targets $y_\tau^{(j)}$ for $-d + 1 \leq \tau < t$, the number of times $a$ (call this $N_t(a)$) has been selected up to time $t - 1$, and the reward noise variables $\varepsilon_\tau$ for $1 \leq \tau < t$, such that $a_\tau = a$. This is because $\left\langle w_t^{(j)}, x_a \right\rangle = \frac{1}{N_t(a)}\sum_{\tau = -d+1}^{t-1} \mathbf{1}_{a_\tau = a} y_\tau^{(j)}$ in the case where all the $x_a$ are orthogonal.

Consider the following procedure: sample $TA(1 + M)$ independent random variables ahead of time: $\eta_{\tau,a}$, an i.i.d copy of $\eta_t$ for each $\tau \in [T], a \in [A]$, and $y_{\tau,a}^{(j)} \sim \mathcal{N}(0,1)$ for each $\tau \in [T], a \in [A], j \in [M]$. Then, run a deterministic version of Algorithm 1, which at time $t$ uses these $y_{\tau,a}^{(j)} : \tau = 1, \ldots, N_t(a)$ as the random regression targets to compute $\mathrm{bonus}_t(a)$, and observes reward $\langle x_{a_t}, \theta^\star \rangle + \eta_{N_t(a_t)+1, a_t}$. Then, for any $\theta^*$, the distribution of action sequences taken by this procedure is identical to that of Algorithm 1 with sequentially sampled $\eta_t$ and $y_t^{(j)}$, and Lemma 6 holds. By Lemma 5, with probability at least $1 - \delta'$, we have that for any $t \in [T], a \in [A]$,

$$\max_{j \in [M]} \left|\left\langle w_t^{(j)}, x_{t,a} \right\rangle\right| \leq c_2 \sqrt{\frac{1}{\lambda + N_t(a)}}\sqrt{\log M + \log(AT/\delta')},$$

and for $a^*$,

$$c_1 \sqrt{\frac{1}{\lambda + N_t(a)}}\sqrt{\log M - \log\log(T/\delta')} \leq \max_{j \in [M]} \left|\left\langle w_t^{(j)}, x_{t,a^*} \right\rangle\right|,$$

for absolute constants $c_1, c_2 > 0$, so that the conditions of Lemma 7 hold with the same choice of $\bar{\beta}$. Selecting $\delta' = \delta/(2AT)$, we have by the union bound that this condition holds for all $t \in [T], a \in [A]$, and the proof reduces to that of Proposition 2, with an identical regret bound of

$$R(T) \leq O\left(\sigma A\sqrt{T\log A}\log(T/\delta)\log^+\left(\frac{TBW}{\sigma A}\right)\right).$$

Note here that $A = d$, and that $T \geq A$ is necessary for a non-vacuous regret bound, so that $\log A \leq O(\log T)$ factors in the main paper are suppressed by the $\tilde{O}(\cdot)$ notation. $\qquad\square$

## C  LAZILY RE-RANDOMIZED ACB

---
**Algorithm 3** Lazy-ACB for fixed action linear bandits
---
**Parameters:** ensemble size $M$; $\beta, \gamma, \lambda$, fixed action set $\mathcal{A}$
1: For all $i = 1, \ldots, d$, set warm-start features $a_{-i+1} = \sqrt{\lambda} e_i$
2: Set $\tau = 1$
3: Set $\omega = 0$
4: **for** $t = 1, \cdots, T$: **do**
5:     **if** $\omega = 0$ **then**
6:        Update parameters:

$$\hat{\theta}_t := \arg\min_{\theta} \sum_{i=1}^{t-1} (\langle a_i, \theta \rangle - r_i(a_i))^2 + \lambda \|\theta\|^2$$

7:        Sample new targets $y_{t'}^{(j)} \sim \mathcal{N}(0,1)$ for each $j \in [M], t' \in \{-d+1, \ldots, t-1\}$
8:        Update bonus ensemble for each $j \in [M]$,

$$w_t^{(j)} := \arg\min_{w} \sum_{t'=-d+1}^{t-1} \left( \langle a_{t'}, w \rangle - y_{t'}^{(j)} \right)^2$$

9:        Compute per-arm bonuses $\text{bonus}_t(a) := \beta \cdot \max_{j \in [M]} \left| \left\langle a, w_\tau^{(j)} \right\rangle \right|$ for each $a \in \mathcal{A}$
10:       Set $s_t := \arg\max_{a \in [A]} \langle a, \hat{\theta}_t \rangle + \text{bonus}_t(a)$
11:       Set $\omega = \left\lceil \frac{\gamma}{\text{bonus}^2(s_t)} \right\rceil$
12:       $\tau = t$
13:     Choose action $a_t = a_\tau$
14:     Observe reward $r_t(a_t) = \langle s_t, \theta^* \rangle + \varepsilon_t$
15:     Decrease $\omega$ by 1
---

In this section, we present the algorithm (see Algorithm 3) and regret analysis of a lazy-version of ACB which only rarely re-randomizes history.

### C.1  USEFUL LEMMAS

**Lemma 8** (Ensemble Bounds). *For $M = \log(2AT/\delta)$, at all updates $\tau$, with probability $1 - \delta$, for all $a \in \mathcal{A}$,*

$$\beta \|a\|_{\tilde{\Sigma}_t^{-1}} \sqrt{\frac{\pi \log 2}{2}} \leq \text{bonus}_\tau(a) \leq \beta \|a\|_{\tilde{\Sigma}_t^{-1}} \sqrt{8 \log \left( \frac{2AT}{\delta} \right)}.$$

*Proof.* Follows by applying Lemma 5 and taking union bound over $\mathcal{A}$ and $t \leq T$. $\qquad\square$

**Lemma 9.** *Let $\tau$ be the time of an update and $\gamma = \frac{\beta^2 \pi \log 2}{2}$. Then, for $\tau' = \tau + \left\lceil \frac{\gamma}{\text{bonus}_\tau^2(a_\tau)} \right\rceil$,*

$$\frac{\det(\tilde{\Sigma}_{\tau'})}{\det(\tilde{\Sigma}_\tau)} \geq \sqrt{1 + \frac{\pi \log 2}{16 \log(2AT/\delta)}}.$$

*Proof.* Note that from $\tau$ to $\tau'$, the action taken remains the same. For the upper bound, using Lemma 11 from Abbasi-Yadkori et al. (2011) and some elementary calculus:

$$
\begin{aligned}
2\log\left(\frac{\det(\tilde{\Sigma}_{\tau'})}{\det(\tilde{\Sigma}_{\tau})}\right) &\geq \sum_{i=1}^{\tau'-\tau} \|a_\tau\|_{\tilde{\Sigma}_{\tau+i-1}^{-1}}^2 \\
&= \sum_{i=1}^{\tau'-\tau} a_\tau^T\left(\tilde{\Sigma}_\tau + (i-1)a_\tau a_\tau^T\right)^{-1} a_\tau \\
&= \sum_{i=1}^{\tau'-\tau} \|a_\tau\|_{\tilde{\Sigma}_\tau^{-1}}^2 - \frac{\|a_\tau\|_{\tilde{\Sigma}_\tau^{-1}}^4}{1/(i-1) + \|a_\tau\|_{\tilde{\Sigma}_\tau^{-1}}^2} \\
&= \sum_{i=1}^{\tau'-\tau} \frac{\|a_\tau\|_{\tilde{\Sigma}_\tau^{-1}}^2}{1 + (i-1)\|a_\tau\|_{\tilde{\Sigma}_\tau^{-1}}^2} \\
&\geq \int_{1/\|a_\tau\|_{\tilde{\Sigma}_\tau^{-1}}^2}^{1/\|a_\tau\|_{\tilde{\Sigma}_\tau^{-1}}^2 + (\tau'-\tau)} \frac{1}{u}du \\
&\geq \log\left(1 + (\tau'-\tau)\|a_\tau\|_{\tilde{\Sigma}_\tau^{-1}}^2\right) \\
&\geq \log\left(1 + \frac{\gamma}{8\beta^2\log(2AT/\delta)}\right) \\
&= \log\left(1 + \frac{\pi\log 2}{16\log(2AT/\delta)}\right).
\end{aligned}
$$

$\square$

**Lemma 10.** *Let $\tau_t$ be the largest index $\leq t$ when the update happened and $\gamma = \frac{\beta^2\pi\log 2}{2}$, then*

$$
\frac{\det(\tilde{\Sigma}_t)}{\det(\tilde{\Sigma}_{\tau_t})} \leq 2e.
$$

*Proof.* Suppose $t = \tau_t$, then the bound trivially holds, thus WLOG, we assume $t > \tau_t$. By our algorithm, we know that, $0 < t - \tau_t \leq \left\lceil \frac{\gamma}{\text{bonus}_{\tau_t}^2(a_\tau)} \right\rceil$. This implies, $\text{bonus}_\tau^2(a_\tau) \leq \gamma$. Now, using Lemma 11 from Abbasi-Yadkori et al. (2011) and some elementary calculus:

$$
\begin{aligned}
\log\left(\frac{\det(\tilde{\Sigma}_t)}{\det(\tilde{\Sigma}_{\tau_t})}\right) &\leq \sum_{i=1}^{t-\tau_t} \|a_{\tau_t}\|_{\tilde{\Sigma}_{\tau+i-1}^{-1}}^2 \\
&= \sum_{i=1}^{t-\tau} \frac{\|a_{\tau_t}\|_{\tilde{\Sigma}_{\tau_t}^{-1}}^2}{1 + (i-1)\|a_{\tau_t}\|_{\tilde{\Sigma}_{\tau_t}^{-1}}^2} \\
&\leq \|a_{\tau_t}\|_{\tilde{\Sigma}_{\tau_t}^{-1}}^2 + \int_{1/\|a_{\tau_t}\|_{\tilde{\Sigma}_{\tau_t}^{-1}}^2}^{1/\|a_{\tau_t}\|_{\tilde{\Sigma}_{\tau_t}^{-1}}^2 + t-\tau-1} \frac{1}{u}du \\
&\leq \|a_{\tau_t}\|_{\tilde{\Sigma}_{\tau_t}^{-1}}^2 + \log\left(1 + (t-\tau-1)\|a_{\tau_t}\|_{\tilde{\Sigma}_{\tau_t}^{-1}}^2\right) \\
&\leq \frac{2\text{bonus}_{\tau_t}^2(a_{\tau_t})}{\beta^2\pi\log 2} + \log\left(1 + \frac{2\gamma}{\beta^2\pi\log 2}\right) \\
&\leq \frac{2\gamma}{\beta^2\pi\log 2} + \log\left(1 + \frac{2\gamma}{\beta^2\pi\log 2}\right) \\
&= 2e.
\end{aligned}
$$

$\square$

## C.2 REGRET BOUND

**Theorem 4** (ACB regret bound with lazy updates)**.** *Algorithm 3, with lazy re-randomized history, ensemble size* $M = \lceil \log(AT/\delta) \rceil$*,* $\lambda = \sigma^2/W^2$*, and* $\beta = \left( \sqrt{\lambda} W + \sqrt{2} \sigma \sqrt{d \log\left(1 + \frac{TB}{\lambda d}\right) + \log(T/\delta)} \right) \sqrt{\frac{2}{\pi \log 2}}$*, and* $\gamma = \frac{\beta^2 \pi \log 2}{2}$*, obtains a regret bound of*

$$R(T) \leq \tilde{O}(\sigma d \sqrt{T \log A}) \log(T/\delta) \log^+ \left( \frac{TBW}{\sigma d} \right)$$

*in the fixed action linear bandit setting, with probability at least* $1 - \delta$*. The algorithm re-randomizes at most* $O(d \log^+(TBW/\sigma d) \log(AT/\delta))$ *times.*

*Proof.* Let $\tau_t$ be the largest index $\leq t$ when the update happened, then

$$
\begin{aligned}
\mathrm{regret}_t &= \langle a^*, \theta^* \rangle - \langle a_{\tau_t}, \theta^* \rangle \\
&\leq \left\langle a^*, \hat{\theta}_{\tau_t} \right\rangle + \beta_{\tau_t} \|a^*\|_{\tilde{\Sigma}_{\tau_t}^{-1}} - \langle a_{\tau_t}, \theta^* \rangle \\
&\leq \left\langle a_{\tau_t}, \hat{\theta}_{\tau_t} \right\rangle + \mathrm{bonus}_{\tau_t}(a_{\tau_t}) - \mathrm{bonus}_{\tau_t}(a^*) + \beta_t \|a^*\|_{\tilde{\Sigma}_{\tau_t}^{-1}} - \langle a_{\tau_t}, \theta^* \rangle \\
&\leq \mathrm{bonus}_{\tau_t}(a_{\tau_t}) - \mathrm{bonus}_{\tau_t}(a^*) + \beta_{\tau_t} \|a^*\|_{\tilde{\Sigma}_{\tau_t}^{-1}} + \beta_{\tau_t} \|a_{\tau_t}\|_{\tilde{\Sigma}_{\tau_t}^{-1}} \\
&\leq \left( \beta \sqrt{8 \log\left(\frac{2AT}{\delta}\right)} + \beta_{\tau_t} \right) \|a_{\tau_t}\|_{\tilde{\Sigma}_{\tau_t}^{-1}} + \left( \beta_t - \beta \sqrt{\frac{\pi \log 2}{2}} \right) \|a^*\|_{\tilde{\Sigma}_{\tau_t}^{-1}} \\
&\leq \left( \beta \sqrt{8 \log\left(\frac{2AT}{\delta}\right)} + \beta_{\tau_t} \right) \|a_t\|_{\tilde{\Sigma}_t^{-1}} \frac{\det(\tilde{\Sigma}_t)}{\det(\tilde{\Sigma}_{\tau_t})} + \left( \beta_t - \beta \sqrt{\frac{\pi \log 2}{2}} \right) \|a^*\|_{\tilde{\Sigma}_{\tau_t}^{-1}} \\
&\leq 2e \left( \beta \sqrt{8 \log\left(\frac{2AT}{\delta}\right)} + \beta_{\tau_t} \right) \|a_t\|_{\tilde{\Sigma}_t^{-1}} + \left( \beta_t - \beta \sqrt{\frac{\pi \log 2}{2}} \right) \|a^*\|_{\tilde{\Sigma}_{\tau_t}^{-1}}
\end{aligned}
$$

where the first inequality comes from the validity of the confidence bound for $a^*$, the second from the fact that $a_{\tau_t} = \arg\max_a \left\langle a, \hat{\theta}_{\tau_t} \right\rangle + \mathrm{bonus}_{\tau_t}(a)$, the third from the validity of the confidence bound for $a_{\tau_t}$, the fourth follows from Lemma 8, the fifth follows from Lemma 12 Abbasi-Yadkori et al. (2011) and lastly the sixth follows from Lemma 10.

Setting $\delta_t = \delta/T$ we have $\beta_{\tau_t} \leq \beta \sqrt{\frac{\pi \log 2}{2}}$, therefore,

$$\mathrm{regret}_t \leq 2e \left( \sqrt{8 \log\left(\frac{2AT}{\delta}\right)} + \sqrt{\frac{\pi \log 2}{2}} \right) \beta \|a_t\|_{\tilde{\Sigma}_t^{-1}}.$$

Using the above, we have

$$
\begin{aligned}
R = \sum_{t=1}^{T} r_t &\leq \sqrt{T \sum_{t=1}^{T} r_t^2} \leq 2e \left( \sqrt{8 \log\left(\frac{2AT}{\delta}\right)} + \sqrt{\frac{\pi \log 2}{2}} \right) \beta \sqrt{2T \log(\det(\tilde{\Sigma}_T) \lambda^{-d})} \\
&\leq 2e \left( \sqrt{8 \log\left(\frac{2AT}{\delta}\right)} + \sqrt{\frac{\pi \log 2}{2}} \right) \beta \sqrt{2Td \log(1 + TB/\lambda d)}.
\end{aligned}
$$

Lastly, we need to bound the number of times we refresh randomness. By Lemma 9, each time we refresh, we increase $\det(\tilde{\Sigma}_t)$ by a factor of $\sqrt{1 + \frac{\pi \log 2}{16 \log(2AT/\delta)}}$. Since $\det(\tilde{\Sigma}_t)/\det(\lambda I)$ is bounded by $(1 + TB/\lambda d)^d$, the number of times we refresh is $k = O(d \log^+(TBW/\sigma d) \log(AT/\delta))$. Formally, $k$ must satisfy $\left( 1 + \frac{\pi \log 2}{16 \log(2AT/\delta)} \right)^{k/2} = \Theta\left( (1 + TB/\lambda d)^d \right)$. $\qquad\square$

