# OpenReview forum: "Anti-Concentrated Confidence Bonuses For Scalable Exploration"
_ICLR.cc/2022/Conference — ICLR 2022 Poster_

### Official Review · Reviewer_AKDp · 2021-10-31

**Correctness:** 2
**Technical Novelty And Significance:** 3
**Empirical Novelty And Significance:** 2
**Recommendation:** 5
**Confidence:** 3

**Main Review:**

Strengths:

This paper provides a rigorous analysis of how generalized exploration bonuses for deep RL can be derived from fundamental exploration bonuses from linear stochastic bandits. Tackling the scalability issue via bypassing covariance matrix inversion by taking a maximum over an ensemble of regressors is a novel contribution.

The proposed algorithms come with different update frequencies ranging from always re-randomizing to incremental updates. Regret bounds that are analogous to that of LinUCB are derived for different update frequencies for the case of linear stochastic bandits. Having a theoretical foundation based on linear stochastic bandits supports the practical contribution on the deep RL side.

The main contribution of this paper comes from applying ACB principle for exploration in deep RL where high-dimensionality of the policy network prohibits efficient covariance matrix inversion.

Weaknesses:

The main selling point of this paper is a computationally efficient procedure for exploration bonus calculation that does not require costly covariance matrix inversion. A major concern related to this is that the computational efficiency of the proposed approach is not analyzed rigorously. The advantage of the proposed method compared to performing rank-1 updates in the implementation of LinUCB is not articulated well. The paper does not provide any empirical evidence that supports such a performance gain.

It is also not clear how the current work improves upon exploration schemes proposed in Kveton et al. (2019a) and Ishfaq et al. (2021). While it is claimed that they resemble the always re-randomizing variant of ACB, it is not clear why can’t they be turned into something in line with lazily-rerandomizing and never-rerandomizing variants with simple algorithmic tweaks as done in this paper. It will also be good to provide an empirical comparison of Algorithm 1 with these algorithms for the multi-armed bandit setting.

Algorithm 1 requires recursive least-squares optimization oracles. The per-iteration computational cost of implementing should also be discussed in the main paper.

Simulations do not show a clear advantage for ACB over RND in terms of visiting new states and maximizing extrinsic rewards. It is not clear why one would not resort to other state-of-the-art methods for exploration in deep RL. In addition, simulations do not demonstrate the computational savings induced by ACB. The paper claims that its main contribution is a novel exploration bonus for deep RL. But it does not provide enough evidence about why one would prefer ACB over other deep RL exploration bonuses.

Kveton et al. 2019 – double citation.

Synthetic bandit experiments should involve comparisons with other benchmarks as well as LinUCB.


**Summary Of The Paper:**

This paper proposes a novel exploration method called anti-concentrated confidence bounds (ACB) that provably approximates the elliptical exploration bonus of LinUCB by using an ensemble of least-squares regressors. ACB computes elliptical confidence intervals for each action by taking a maximum over the predictions of the ensemble. While doing this, ACB bypasses costly covariance matrix inversion, which can be problematic, especially for high-dimensional problems. It is shown that ACB enjoys near-optimal performance in linear stochastic bandits when the cardinality of the action set is polynomial in the action feature dimension. However, the main contribution of this work comes from extending ACB principle for computing exploration bonuses in deep RL. Comparison of ACB with state-of-the-art deep RL exploration methods on Atari benchmarks demonstrates the competitiveness of the proposed approach.


**Summary Of The Review:**

Overall, this paper proposes an interesting way to scale exploration bonuses from linear stochastic bandits for deep RL. However, the paper does not provide convincing evidence that supports the advantages of using ACB both in bandit and deep RL setups. In particular, computational and performance improvements pertaining ACB are not explained well.

---

> ### Author Response · Authors · 2021-11-22
> **response to reviewer AKDp**
>
> Thanks for your review. We provide several clarifications on the points raised below.
>
> - *Computational savings.* In the paper, we have argued that the heuristic of 1 SGD step per iteration (with $Md$ time complexity per update) is faster than rank-1 updates ($d^2$ time per update). The former, with never-rerandomizing targets, is the method we demonstrate to be scalable and effective in practice. We’ve added a clarifying sentence to the revised manuscript at the bottom of page 6.
>
>     That said, there remains a gap between this algorithm and what we can fully analyze, including the optimization error of one step of SGD in the recursive least squares problem, and the possibly correlated tail events in never-rerandomized bonuses. We leave these as open theoretical questions (with partial progress via the rarely-rerandomizing and MAB analyses). We again stress that the main contribution of this paper is empirical.
>
> - *Comparison with other “random target regression” algorithms.* The reviewer is requesting comparisons with LinUCB and the efficient versions of [Kveton et al. ‘19a, Ishfaq et al. ‘21]. We include a regret plot for LinUCB, which we have placed in Figure 2 (red) in the updated manuscript. Regarding the others: the comparisons are already present between the left and right sides of Figure 2 (The never-rerandomizing version of [Kveton et al. ‘19a] is the M=1 OLS instantiation of ACB, and likewise with [Ishfaq et al. ‘21] for general M). The purpose of the synthetic experiments is not to establish comparisons between regrets, as there seems to be a soft computation time vs. approximation error tradeoff-- it is to demonstrate the sublinear-regret performance of never-rerandomizing variants. Again, this serves as a sanity check towards developing our main contribution (Algorithm 2). Overall, we feel that our paper supplements these papers by providing more clarity regarding the tradeoffs with using noise for exploration, and demonstrates that, with the right design choices, these techniques can lead to competitive performance in deep RL settings.
>
> - *Deep RL comparisons.* We want to emphasize that we do not argue that ACB is empirically higher performing than RND. Instead, we are advocating for ACB as an algorithm that is *competitive* with RND, can serve as a drop-in replacement, and that has the benefit of being theoretically transparent. We feel this work is an important step in bridging theory and practice in the space of exploration bonuses for deep RL.

---

> > ### Comment · Reviewer_AKDp · 2021-11-23
> > **question**
> >
> > Thanks for the response. My concerns about points 1 & 2 are addressed. For point 3, is there any reason other than being “theoretically transparent” that one should prefer ACB over other state-of-the-art deep RL bonuses in practice? For instance, is it possible to say that while empirically, the performance of ACB is close to that of RND for most of the cases, it shows a significant computational advantage? I am not asking for a theoretical analysis of complexity but an empirical demonstration of computational efficiency. In addition, please discuss the merits of being “theoretically transparent” in a deep learning setting. For instance, does it offer any new insights on how exploration works in Atari games?

---

> > > ### Author Response · Authors · 2021-11-23
> > > **re: question**
> > >
> > > As a practical matter, the computational efficiency of ACB is not superior to RND.
> > >
> > > The reviewer appears to be asking us to argue for the merits of theoretical transparency in algorithm design. Like many in machine learning, we believe that the gold standard for “understanding” an approach is an ability to mathematically characterize its behavior in well-understood, fundamental settings (such as linear and multi-armed bandits). Advancing our theoretical understanding of usable intrinsic rewards in deep RL is, on its own, noteworthy progress.
> > >
> > > Separate from this, a theoretically transparent algorithm is one that can offer clear directions for innovation: In our conclusion, we mention some possible avenues for improvement, such as eliminating the regret dependency on $\log(A)$, which we expect would also lead to empirical gains.

---

### Official Review · Reviewer_Ezvk · 2021-11-02

**Correctness:** 4
**Technical Novelty And Significance:** 3
**Empirical Novelty And Significance:** 3
**Recommendation:** 6
**Confidence:** 4

**Main Review:**

Overall, the paper is well-motivated and clear in presentation. The logic flow of the paper is also good: after showing the efficiency problem of LinUCB, the authors provide a theoretically-grounded algorithm for linear bandits with much less computation overhead. Then this algorithm is applied to deep RL algorithms. Although the proposed ACB is not completely novel (as also mentioned in the related work section), the paper is well-motivated and the resultant algorithm is effective. Therefore, I vote for acceptance.

While I’m mostly happy with the paper, I have the following questions/suggestions:

To me, the reason why Alg. 2 works are the following: if we have seen a specific $g$ many times, then $y$s corresponding to this similar $g$s will be normally distributed, hence these are approximately zero-centered. And then while minimizing the loss function, $w$ is incentivized to be orthogonal to such $g$s, making the intrinsic reward small. Given this intuition, I could imagine different ways of setting $y$s could influence the performance significantly. Therefore, although the current experiments are sufficient, it would be nice for the authors to show how different ways of setting $y$ influence the performance.

**Summary Of The Paper:**

This paper proposes a new intrinsic reward method for continuous-action-space deep RL algorithms. The proposed algorithm is inspired by the bonus term of LinUCB. The proposed algorithm improves the efficiency of its reward bound’s computation by using a randomized variant of the original reward bonus. The authors theoretically justified the optimality of this ACB bonus term in the linear case, showing that this is a reasonable replacement of the LinUCB reward bonus. Finally, the authors add the ACB bound to deep RL algorithms. Although this algorithm comes with no theoretical guarantee, experiments show that the ACB intrinsic reward is competitive compared to adopted baselines.

**Summary Of The Review:**

Overall, I tend to vote for acceptance. Although the proposed method shares some similarities with existing approaches, the theoretical part is well-motivated and clearly explained, and the empirical part demonstrates the effectiveness of the proposed algorithm.

---

> ### Author Response · Authors · 2021-11-22
> **response to reviewer Ezvk**
>
> Thanks for the great question! At a high level, Algorithm 2 works for the same reason Algorithm 1 does: the weights of linear regressors trained on Gaussian random noise capture information about the inverse covariance matrix. This can be seen by expanding the squared predictions of a single regressor using the analytical solution for least squares:
>
> $(x^T \hat{w})^2 = x^\top \Sigma^{-1} (\sum_i x_i y_i)(\sum_i x_i y_i)^\top \Sigma^{-1} x = x^\top  \Sigma^{-1}  X YY^\top X^\top \Sigma^{-1}x$, which in expectation is the elliptical potential as long as the noise distribution is isotropic ($\mathbb{E}[YY^\top] = I$). Indeed, many choices of noise could potentially suffice, as long as they have sufficient anti-concentration. In light of the reviewer’s comments, we’ve run ACB using zero mean, unit variance Laplace noise, and demonstrate its functionality for a few Atari environments here (https://imgur.com/a/tDl8ykS). Note that this is a somewhat less natural choice than Gaussian noise, since it’s not spherically symmetric, and without tuning hyperparameters performance is a bit worse than in the Gaussian case. Models are trained only on intrinsic rewards, like in Figure 7.
>
> Interestingly, increasing the variance of the noise is equivalent to increasing the regressor’s learning rate, so it’s worth noting that there’s some interaction between those terms. We would be happy to more thoroughly explain these points and to include the above plots in the revision.

---

### Official Review · Reviewer_1PxN · 2021-11-07

**Correctness:** 4
**Technical Novelty And Significance:** 3
**Empirical Novelty And Significance:** 4
**Recommendation:** 8
**Confidence:** 4

**Main Review:**

The paper investigates an interesting problem, which is highly relevant for the ICLR community. The paper is very well-written and straightforward to follow. It also presents the algorithms and results clearly and precisely. Moreover, most relevant works I am aware of are properly cited.

Although the main contribution of the paper is empirical, the variants of the ACB algorithm admit a solid and novel design. Another positive feature is that the sound empirical performance of ACB when compared to state-of-the-art. All these support ACB as a viable exploration bonus for deep reinforcement learning.

Due to limited time, I was unable to check the proofs. Nonetheless, the results appear correct to me.

Besides some minor, easily-fixable comments listed below, I have the following comment. The reported regret bounds for the ACB variant for linear bandits all match the minimax regret for linear bandits, up to logarithmic factors, assuming that the number of actions grows polynomially with $d$. In my opinion, it would be informative if the paper compares the reported bounds with the best available bound considering $\log(T)$ factors. My quick check reveals that ACB’s regret is worse than the best available regret bound by a multiplicative factor of $O(\log^{3/2}(T))$. Please comment on whether this is correct and whether this is improvable.

Minor:

- As far as I understood, all norms used in the paper are Euclidean norms. If so, stating this explicitly could be helpful.

- p. 5: that that => than that

- p. 7: one reference is inappropriately inserted (see “?”)

- Both $L_2$ and $\ell_2$ are used to indicate the same notion.

- p. 4: In the formula $r_t(a)\le \hat r_t(a) + bonus_t(a)$, unless I am missing something, I think $r_t(a)$ in the left-hand side should be $\langle x_{t,a_t},\theta^*\rangle$.

- p. 13: The value of $\gamma_E$ is missing.


**Summary Of The Paper:**

This paper studies using bonus for guiding exploration in reinforcement learning with large action spaces. The paper focuses on RL algorithms using LinUCB-style exploration bonuses. LinUCB, developed originally for stochastic linear bandits, has been a theoretically successful algorithm for problems admitting a linear reward structure. However, using such bonuses entails computing a matrix inversion, which may not be tractable in high dimensional problems.

The main contribution of the paper is to propose Anti-concentrated Confidence Bounds (ACB) with the aim of efficiently approximating the LinUCB bonus without matrix inversions. The main idea of ACB is that it maintains $M$ linear regressors trained to predict i.i.d. noise drawn from $\mathcal N(0,1)$. The algorithm then set the bonus proportional to the maximum deviation over the regressors from the mean. Two versions are considered: one for linear bandits and one for deep RL. For linear bandits, the authors prove a regret of $\tilde O(d\sqrt{T\log A})$ with high probability under mild assumptions (e.g., logarithmically in $T$ many regressors). The variant for deep RL, however, is examined through numerical experiments where the algorithm is shown to be competitive with state-of-the-art.



**Summary Of The Review:**

This paper presents a novel exploration bonus, called ACB, that can be used to design viable exploration strategies for linear bandits and deep reinforcement learning. ACB admits a solid and novel design, and shows competitive performance with state-of-the-art. Finally, the paper is well-written and well-executed.

---

> ### Author Response · Authors · 2021-11-22
> **response to reviewer 1PxN**
>
> Thanks very much for the thorough review of our work.
>
> - *$\log(T)$ dependences.* We believe our analysis is compatible with a stopping time trick (see https://tor-lattimore.com/downloads/talks/2018/aaai/linear-bandits.pdf, page 44) to save one $\sqrt{\log T}$ factor from the union bound over time. However, we believe the final remaining $\log(T)$ factor, arising from the union bound over upper-tail events for the stochastic bonuses, is inevitable, at least in the always-refreshing case. Obtaining an optimal dependence on $\log T$ factors (perhaps with an algorithm and analysis like [Jamieson et al. ‘13]) is an interesting theoretical question. We note that the more salient suboptimality is the log(A) factor (also present in [Kveton et al. ‘19, Ishfaq et al. ‘21]), which can incur a factor of sqrt(d) if there are $\exp(d)$ actions.
> - *Minor points.* We are very appreciative of the close reading, and have addressed the typos in the recent revision of the manuscript.
>
> [Jamieson et al. ‘13] “lil' UCB : An Optimal Exploration Algorithm for Multi-Armed Bandits”.

---

### Decision · Program_Chairs · 2022-01-20

**Decision:**

Accept (Poster)

**Comment:**

This paper tackles the problem of exploration in Deep RL in settings with a large action space. To this end, the authors introduce an intrinsic reward inspired by the exploration bonus of LinUCB. This novel exploration method called anti-concentrated confidence bounds (ACB) provably approximates the elliptical exploration bonus of LinUCB by using an ensemble of least-squares regressors. This allows ACB to bypass costly covariance matrix inversion, which can be problematic for high-dimensional problems (hence allowing it to be used in large state spaces). Empirical experiments show that ACB enjoys near-optimal performance in linear stochastic bandits. However, experiments on Atari benchmark fail to show any practical advantage of ACB over current methods, neither computation nor performance-wise. That being said, the proposed ACB approach is theoretically transparent, which contributes to advancing our theoretical understanding of usable intrinsic rewards in deep RL and can inform theoretically motivated directions for improvement and further research, while being on par with SOTA. I believe that this makes the contribution of this work strong enough for acceptance.